# Metabolic and Performance Responses to a Simulated Routine in Elite Artistic Swimmers

**DOI:** 10.3390/sports10120190

**Published:** 2022-11-25

**Authors:** David J. Bentley, Eric Viana, Heather M. Logan-Sprenger

**Affiliations:** 1Faculty of Health Science, Ontario Tech University, 2000 Simcoe Street North, Oshawa, ON L1G 0C5, Canada; 2Canadian Sport Institute Ontario, 857 Morningside Avenue, Toronto, ON M1C 0C7, Canada

**Keywords:** fatigue, hypoxia, hypercapnia, performance, aerobic, athletes

## Abstract

The purpose of this investigation was to examine the interrelationship between time spent underwater (UW), movement frequency and accompanying blood acid base balance response. Elite artistic swimmers (n = 6) participated in the investigation and were all familiar with the testing procedures. All athletes completed the same choreographed artistic swimming routine. The routine was videoed and the number of movements during each ‘lap’ of the routine counted. Fingertip capillary blood samples were collected prior to the routine 60 sec post routine for pH, partial pressure of carbon dioxide (pCO2), partial pressure of oxygen (pO2), bicarbonate (HCO_3_^−^) and potassium (K+). and lactate (La) concentration (mmol/L). On a separate day an incremental exercise test to exhaustion was performed on a cycle ergometer for determination of maximal oxygen uptake (VO2max). Over half the routine was performed underwater (56 ± 4%). Aside from pCO2 (−1.07 ± 12.29%, *p* = 0.686), there were significant changes in all variables measured from the BG analysis. VO2peak was significantly correlated to total UW of the routine (r = −0.93; *p* = 0.007). as well as ∆PO2 r = 0.47 and ∆HCO3 r = 0.51. There was also a significant correlation between total UW and post routine pCO2 (r = 0.86; *p* = 0.030). There was also a significant correlation between total movements during the routine and post pO2 (r = −0.83; *p* = 0.044). These data show UW in combination with movement rate during a AS routine imfluence the metabolic response to the exercise. In addition, VO2max represents an important performance variable influencing AS performance.

## 1. Introduction

Artistic swimming (AS) is a dynamic and aesthetic sport characterized by repeated periods of underwater (UW) exposure (apnea) while performing explosive movements [1]. The explosive movement patterns combined with breath holding (BH) evoke physiological responses and fatigue manifestation unique to AS [2,3]. The rapid movements are required to successfully perform ‘elements’, which are specific body positions and patterns incorporated into the athlete’s routine which athletes are scored/judged upon. Anecdotally, athletes record greater scores if they are able to perform more movements in a routine allowing the opportunity to include more scoring opportunities into routine.

It is widely accepted that with increases in exercise intensity there is an increase in non-metabolic CO2 production and elevated oxygen consumption (VO2) to satisfy the energy demands of the given exercise intensity [4]. During severe exercise, the increase in non-metabolic CO2 production results in a metabolic shift towards hypercapnia [5]. Previous research in AS has utilized blood gas analysis to examine changes in pH, HCO_3_^−^, pO2 and pCO2, in which there was a shift towards compensatory metabolic acidification as indicated by reductions in blood pH and HCO_3_^−^ [1,6,7]. These studies in elite AS have also shown the blood lactate response (indicative of a greater anaerobic energy contribution) is influenced by the UW, but also a function of the event (solo vs. team [6,7]. In addition there is a considerable hypercapnic response which is influenced by aerobic fitness level [8]. Performing a greater number of movements stemming from a greater movement rate, may also result in an increased energy cost for a routine [1,6,7]. This is further complicated by the bradycardia response observed during exercise performed in water [1,2] There are no studies have examined specifically how movement frequency combined with the duration of time spend underwater influences the physiological response in AS. The metabolic consequences and fatigue manifestation of UW and movement frequency has not been investigated in elite artistic swimmers. This is important during AS training could be associated with the metabolic (hypercapnic) demand imposed on swimmers and this is especially significant with some reports of ‘black out’ (loss of consciousness during AS training perhaps due to the hypercapnic response [9,10,11]. At the same time there has been no scientific investigation that has examined the changes rate of movement (fatigue) over a AS routine and the physiological factors that influence that fatigue manifestation.

An enhanced aerobic capacity (VO2max) has been frequently cited as an important physiological characteristic of short endurance events such as rowing and cycling [12,13]. Aerobic capacity represents the potential for aerobic energy production which is important for sports such as AS of duration sport 5 min [1]. Therefore, aerobic capacity could be viewed as an important physiological factor influencing performance in AS. However, there has been some controversy as to the importance of maximal oxygen uptake (VO2max) in AS performers potentially due to the means with which this variable is measured (in water or on land) [8,14,15,16].

The purpose of this research was to evaluate the impact of the rate of movement as well as UW (apneic)time on the physiological responses to a simulated AS event in highly trained athletes. This study also examined the relationship between performance and the physiological responses during an AS routine and VO2max in elite competitors.

## 2. Materials and Methods

### 2.1. Participants

Six highly trained artistic swimmers (mean ± SD age 14.2 ± 1.2 years, height 169 ± 4 cm, body mass 55.7 ± 4.9 kg) voluntarily participated in the data collection as part of their routine performance analysis. The participants were competing at a national level and were part of a provincial high performance program with some participating at world junior championship level. The athletes were approached in agreement with the head coach and asked to participate. The criteria for inclusion were that participants’ were >15 years of age and be free of injury and illness. Two athletes out of the squad of 10 were not included due to injury. The athletes were provided with written documentation on the investigation benefits and risks. The participants subsequently gave a voluntary informed consent document to participate in the study which was approved by both the Research Ethics Board (REB) of both the Ontario Technical University and the Canadian Sport Institute Ontario (CSIO) (REB#105).

### 2.2. Experimental Overview

Participants completed a maximal incremental exercise test to exhaustion on a cycle ergometer (Velotron, RacerMate, Inc., Seattle, WA, USA) for determination of maximal aerobic capacity (VO2max). One week after completing the incremental exercise test participants completed their portion of their Team ‘Highlight’ routine during a practice session prior to competition. Blood samples were obtained before and after the routine for blood gas analysis. In addition, the routine was video recorded for subsequent analysis of movement count to determine rate of movement frequency.

### 2.3. Incremental Bike Test

Participants completed the incremental exercise test on a stationary cycle ergometer (Velotron, RaceMate Inc., Seattle, WA, USA). equipped with a power meter (SRM). Each participant performed a 5 min warm-up at 0.5 watts per kilogram (kg) of total body mass (w/kg). The test itself commenced with 3 min submaximal stages at 50, 100 and 150 watts (W) following three initial workloads the required power was increased by 15 W every 30 sec to exhaustion. Expired gasses were continuously measured breath by breath using a mask connected to a one way valve which fed expired air through a pneumotach, oxygen and carbon dioxide sensor which then fed real time data (Moxus, AEI Technologies, Bastrop, TX, USA). Heart rate (HR) was measured using a HR belt linked wirelessly to the metabolic system. The incremental test was terminated when the participant reached volitional fatigue, and VO2max value was selected after a 15 s plateau in relative VO2 and respiratory exchange ratio (RER) > 1.00.

### 2.4. Simulated Swimming Routine

After 7 days, participants completed their portion of an AS team event. Each participant completed a 600 m warm up consisting of the freestyle stroke, sculling and select elements. Upon completion of the warm up participants exited the 25 m pool and were fitted with water resistant HR monitor (Polar OH1, Polar Electro Oy, Kempele, Finland). The HR monitors were secured in place with a neoprene compression sleeve around the mid portion of the bicep as recommended by the manufacturers. Participants then provided a fingertip capillary blood gas (BG) sample for analysis. The participants index finger was punctured with a single use lancet after being sanitized with an alcohol swab. An 85 µL sample of blood was collected in a 105 µL capillary tube. The 85µL capillary blood samples were immediately analysed using a blood gas analyzer (ABL 80, Radiometer, Copenhagen, Denmark) where the following parameters were measured: pH, partial pressure of carbon dioxide (pCO2), partial pressure of oxygen (pO2), bicarbonate (HCO_3_^−^) and potassium (K+). Blood lactate (mM) was collected post routine from a capillary fingertip puncture site.

Participants then completed their portion of a Team Highlight routine (~2:45 min:seconds) where HR was monitored continuously. The routine was also recorded using a tripod mounted iPad that panned across the length of the pool to ensure the participant was centered in the frame. Upon completion of their routine participants were instructed not to breathe until they had reached the pool side (which took less than 10 sec) and were immediately fitted to a portable gas analyzer (CosMed K4b2, CosMed, Italy) which measured expired gasses breath by breath. Data from the CosMed K4b2 were collected for a minimum of 20 s and peak oxygen uptake (VO2) determined using the reverse extrapolation technique of the oxygen recovery curve [17]

Predicted VO2 was determined using the following formula:pVO2(t) = VO2(t)∙HRend exercise/HR(t)
where pVO2 is the predicted VO2 value at a iven time point of the routine, t represents time expressed as a percentage of the total routine, VO2(t) is the VO2peak value measured in the >20 s the participant was breathing into the CosMed K4b2, and HR is heart rate.

The video footage was separated into four ‘laps’, where a new lap is defined as a change in the direction the participants were swimming. It is similar to a ‘length’ however the term ‘lap’ was selected as the participants never completed a full ‘length’ of the pool’. During each lap movements, movement rate and apneic time (sec) expressed both as absolute and relative values were measured. A movement was defined as: a definitive change in the position or direction of the upper and/or lower limbs as per the element, figure, hybrid figure or strokes and propulsion techniques mandated by the choreography of the routine. An apneic exposure was defined as any event in which the participants face was visibly below the water and was measured with a handheld stopwatch to the nearest tenth of a second.

### 2.5. Statistical Analysis

All statistics were calculated using IBM SPSS Version 25. Mean and standard deviation [95% confidence interval (CI)] were calculated for each variable during the incremental exercise and simulated AS routine. One way analysis of variant (ANOVA) determined the difference in variables before and after the artistic swim routine. Pearson product correlation coefficients were used to examine the relationship between blood and performance parameters in the AS routine and incremental exercise test.

## 3. Results

### 3.1. Artistic Swim Performance Characteristics

Approximately half the AS routine was spent in UW (56 ± 4%). Table 1 shows the key performance variables measured during the AS routine. The relative amount of UW per lap was similar between laps 1 to 3 (57 ± 11%, 52 ± 7%, 55 ± 7%, respectively). However, there was an increase in UW in lap 4 (65 ± 12%). However, this change was not significantly different (t = −1.232, df-10, *p* = 0.246, [CI95: −22.438, 6.458]). The movements per/lap were similar in lap 1 one and 3 (56 ± 16 movements vs. 56 ± 13 movements, respectively), but increased in lap 2 two (62 ± 16), and decreased in lap four 4 (42 ± 19). Lap 1 one showed the greatest movement rate (103 ± 27 no/min) with a reduction in rate in laps 2–4. (lap2: 87 ± 19, lap3: 82 ± 27 and lap4: 87 ± 29, respectively).

### 3.2. Physiological Responses to the Artistic Swim Routine

With the exception of pCO2 (−1.07 ± 12.29%, t = 0.43, df = 6, *p* = 0.686), there were significant changes in all BG parameters during the AS routine (See Table 2). Blood pH decreased by 3.54 ± 0.88% (t = 10.167, df = 5, *p* < 0.000), pO2 increased by 94.5 ± 4.4% (t = −8.16, df = 5, *p* < 0.000), HCO_3_^−^ decreased 47.5 ± 3.1% (t = 19.75, df = 5, *p* < 0.000) and K+ increased by 92.9 ± 19.4% (t = −14.47, df = 5, *p* < 0.000). VO2peak obtained during the incremental cycle test did not significantly differ from VO2peak obtained during the AS routine (53 ± 5 vs. 50 ± 1 mL∙kg∙min^−1^) (t = −2.14, df = 5, *p* = 0.086).

### 3.3. Correlation Coefficients

There was a significant correlation between VO2peak and total UW of the routine (r = −0.93; *p* = 0.007). as well as ∆PO2 r = 0.47 and ∆HCO3 r = 0.51. There was also a significant correlation between total UW and post routine pCO2 (r = 0.86; *p* = 0.030). The UW relative to total routine time (%) was significantly correlated to change (∆) pO2 (r = 0.82; *p* = 0.04). There was also a significant relationship between total movements during the routine and post pO2 (r = −0.83; *p* = 0.044) and movement rate to post HCO_3_^−^ (r = 0.82; *p* = 0.048). There was no significant relationship between the number of movements/lap and % of the lap spent underwater.

## 4. Discussion

The purpose of this study was to examine the concentration of O2/CO2 in the blood and the link with performance based parameters (including movement frequency) as well as the duration of UW exercise, to characteristise the demands and fatigue evoked during AS. We found that the time spent underwater (UWST) and the movement frequency was related to changes in the O2 and CO2 content of the blood. Extending previous findings, this study demonstrates that the time spent underwater (UW) is significantly correlated to the changes in blood parameters namely CO2 production. Furthermore, these findings indicate that firstly VO2peak measured during a land-based protocol using a cycle ergometer is not different to VO2peak obtained upon completion of a simulated AS event. In addition, VO2peak measured during cycling is significantly correlated to the UW as well as the production of CO2 post routine. Contrary to our hypothesis, movement rate during simulated AS solo routine was not significantly related to VO2peak. Interestingly we found that movement frequency was correlated to the change in blood O2 content. This seems to indicate that the greater the oxygen availability in the body the better the swimmers were able to perform the movements in their routine.

The physiological response to AS is unique in that it comprises rapid movements combined with lengthy periods of time underwater. The average apneic exposure shown in this study was similar to other studies [2,3] with 6.21 ± 1.11 s observed in this study vs. 6.8 s [2,3]. The significant changes in both blood pH and HCO_3_^−^ shown in this study indicate a degree of metabolic acidification caused by the increase in non-metabolic CO2 production which is consistent with previous studies in solo and team AS events [6,7]. In this study we have also shown that the length of UW reflects the increase in metabolic acidosis which is an important finding relevant to training demands and subsequent training load manipulations. Training load monitoring in land based sports typically reflects the changes in duration and intensity of individual and successive exercise bouts [18]. There are limited studies examining the process and outcomes of training load on performance optimsation in AS [19,20,21]. However, the additional demand in AS is for athletes to spend a significant time UW which may impose unique chronic adaptations (or maladaptation) to training. Some studies show profound changes in central neurophysiological function with repeated hypoxia exposure [22]. The implication of these and the current findings relate to frequent training exposures in apnea of AS athletes with excessive increases in non-metabolic carbon dioxide (CO2) or hypoxia having deleterious effects on central/cognitive function, and thus decision making, and could contribute to errors and risk of injury [9,11,23,24]. With this in mind, there have been anecdotal reports of underwater blackouts during AS practices which could be due in part to the hypercapnic response due to repeated heavy exercise underwater [10]. Therefore, these findings also are relevant to potential adverse incidences of loss of consciousness in AS athletes [10]. However, the relevance of these findings during longer term periods of training and the implications for training load manipulations is beyond the scope of this study but worthy of future investigations.

The importance of VO2peak is observed in traditional endurance events such as middle and long distance running and road cycling amongst others [13]. Maximal oxygen uptake is also important in shorter events in the vicinity five minutes for example rowing and track cycling [10]. Typical AS routines comprises rapid movements over a period of 4 to 5 min. In the study we found that VO2peak was related to the length of UW exposure. In addition VO2peak is also related to a number of blood related parameters reflecting the metabolic status of the athlete during the routine. The very limited data available examining the significance of VO2peak in artistic swimming have found significant relationships between solo performances [15,16] as well as high levels in elite performing swimmers compared to lower level counterparts [8,15]. Data from this present study supports these findings, and indicates that a more favorable metabolic environment occurs in athletes with a higher aerobic capacity. The significant relationship between VO2peak and total UW of the routine may also indicate that AS athletes with greater aerobic capacity could be better able to perform routines with greater amounts of time UW than those athletes with a lesser aerobic capacity. The observation that an increase in movement rate and total number of movements may require a greater degree of aerobic fitness is an important consideration for athlete preparation and performance analysis. This is important consideration for the preparation of AS and the planning of their training. In addition to the findings the study supports the idea that VO2max during land based test is similar to that in a simulated artistic event and can be used as a surrogate measure of AS performance [8].

This is the first study in elite AS athletes to examine the relationship between performance-based parameters such as movement frequency and the metabolic responses to a simulated competitive AS routine. The subjects recruited for this investigation were small (n = 6) which is a limitation of the study. However, these athletes represent an elite population. Future studies should aim to compare different subgroups of AS athletes (elite vs. recreational). However, the complexity of the sport in terms of a choreographed performance make this difficult. This study shows the metabolic demands and has implications for long term training adaptation, this study did no quantify adaptations to UW exercise. This would be an interesting extension of this work that has implications for talent identification and optimising training in AS. A further limitation of the study is the nature of the AS routine which was a ‘simulated’ competition which may not necessarily reflect the demands of actual competition events.

## 5. Conclusions

The results of this study complement previous studies in actual and simulated AS competitions. This study confirms the unique underwater and metabolic environment that artistic swimmers must endure in order to perform their chosen routine. The results of this study lead to a better understanding of the physiological response during an AS routine which has implications for training and preparation of AS. Specifically the severity of UW swimming and movement rate lead to a higher metabolic demand of the routine which is important for training load, monitoring of adverse responses to AS and strategies for performance enhancement. Future studies are required to examine the long term changes in metabolic response and performance parameters during periods of training in AS.

## Figures and Tables

**Table 1 sports-10-00190-t001:** Performance variables in each ‘lap’ of the simulated artistic swim routine.

Variables	Lap 1	Lap 2	Lap 3	Lap4
Underwater time (UW) (%)	57 ± 11	52 ± 7	55 ± 7	65 ± 12
Movements per lap (no)	56 ± 16	62 ± 16 *	56 ± 13	42 ± 19 *
Movement rate (no/min)	103 ± 27	87 ± 19 *	82 ± 27 *	87 ± 29 *

* Significantly different from lap 1 (*p* < 0.05).

**Table 2 sports-10-00190-t002:** Physiological responses to the simulated artistic swim routine.

Variable	Pre	Post	% Change
Blood lactate (mM)	1.1 ± 0.2	11.5 ± 1.09	90.4
pH	7.42 ± 0.03	7.16 ± 0.04 *	3.59
PO2	68.7 ± 7.70	95.9 ± 5.67 *	39.5
PCO2	37.0 ± 3.48	35.9 ± 2.18	2.97
HCO_3_^−^	23.8 ± 1.63	12.4 ± 1.41 *	−48
K^+^	15.3 ± 1.61	29.0 ± 2. 39 *	89.6

* Significantly different from pre to post exercise (*p* < 0.001).

## Data Availability

The original contributions presented in the study are included in the article. Any inquiries on data specifics can be directed to the corresponding author.

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
