# Peer review of "Metabolic and Performance Responses to a Simulated Routine in Elite Artistic Swimmers"

_sports, 2022, doi:10.3390/sports10120190_

Round 1

Reviewer 1 Report

Line 8 – abbreviation should be in full form at its first use

I have serious concerns regarding the sample size. Only 6 participants were in the study. Was the sample size calculated before data collection

More detail about the participant's recruitment is required. How many participants were invited? From were the participants recruited? What are the inclusion/exclusion criteria? How many participants were excluded based on these criteria?

Line 73 – what are those values

What was the design of the study?

Line 104 – after how many days

Was a posttest taken in the study? If yes, in how many days?

Where are the pre and post-performance data?

The discussion is incomplete. The authors not following the basic format of the discussion. All the findings of the study should be discussed.

What are the limitations of the study? 

Author Response

We thank the reviewers for the time they have allocated to reading and commenting on the manuscript. The comments raised have improved the quality of manuscript and has identified potential opportunities for new lines of research. Below shows each of the reviewers' comments in bold and our replies. Any modifications in the manuscript have been highlighted in red. Any additional text also is seen below under each respective comment.

Review 1

Line 8 – abbreviation should be in full form at its first use

Abbreviation has been included as ‘time spent underwater (UW)

I have serious concerns regarding the sample size. Only 6 participants were in the study. Was the sample size calculated before data collection

The authors understand the reservations around small Sample sizes using traditional parametric statistics However studies in sport nowadays investigate elite populations which by definition small sample sizes. We have mentioned the small sample size as a limitation in the discussion section.

More detail about the participant's recruitment is required. How many participants were invited? From were the participants recruited? What are the inclusion/exclusion criteria? How many participants were excluded based on these criteria?

Further details have been included in the methodology section about the recruitment of participants. The 6 athletes represented the majority of the high performance squad. Others who did not participate were either competing or injured.

The subjects were approached in agreement with the head coach and asked to participate.

Line 73 – what are those values

The sentence has been modified to:

(mean± SD age 14.2 ± 1.2 years, height 169 ± 4 cm,  body mass 55.7 ± 4.9 kg)

What was the design of the study?

The design of the study was a single group case study analysis using correlation statistics to examine the relationship between changes in blood markers and performance indicators during an artistic swimming event. Details of the tests completed, and analysis made are in section 2.2 Experimental overview

Line 104 – after how many days

The AS routine was performed 7 days after the incremental exercise test was performed (see 2.2 experimental overview)

Was a posttest taken in the study? If yes, in how many days?

Posttest (artistic swimming routine) blood samples were obtained. This is not an interventional study examining effective training for example on artistic swimming performance but rather to examine acute responses before and after the artistic swimming routine

Where are the pre and post-performance data?

Full pre and post test (AS routine) are shown in table 2

The discussion is incomplete. The authors not following the basic format of the discussion. All the findings of the study should be discussed.

We have reviewed the discussion section. We have acknowledged the comments raised by the author in terms of structure of the discussion. The basic findings (results) are listed commencing line 12 of the discussion section

‘We found that the time spent underwater (UWST), and the movement frequency was related to changes in the O2 and CO2 content of the blood. Extending previous findings, this study demonstrates that the time spent underwater (UWT) is significantly correlated to the changes in blood parameters namely CO2 production. Furthermore, these findings indicate that firstly VO2peak measured during a land-based protocol using a cycle ergometer is not different to VO2peak obtained upon completion of a simulated AS event. In addition, VO2peak measured during cycling is significantly correlated to the UWT as well as the production of CO2 post routine. Contrary to our hypothesis, movement rate during simulated AS solo routine was not significantly related to VO2peak. Interestingly we found that movement frequency was correlated to the change in blood O2 content.’

We are happy to make modifications based upon any specific incomplete text as the author recommends. Generally speaking, this author uses the format whereby the key results to study are highlighted in the early stages of the discussion and are subsequently focused on in subsequent paragraphs. 

What are the limitations of the study?

Thank you for identifying this potential modification. Text has been added at the end of the discussion:

‘This is the first study in elite AS athletes to examine the relationship between performance-based parameters such as movement frequency and the metabolic responses to a simulated competitive AS routine. The subjects recruited for this investigation were small (n=6) which is a limitation of the study. However, these athletes represent an elite population. Future studies should aim to compare different subgroups of AS athletes (elite vs recreational). However, the complexity of the sport in terms of a choreographed performance make this difficult. This study shows the metabolic demands and has implications for long term training adaptation, this study did not qualify adaptations to UW exercise. This would be an interesting extension of this work that has implications for talent identification and optimising training in AS. A further limitation of the study is the nature of the AS routine which was a ‘simulated’ competition which may not necessarily reflect the demands of actual competition events.’

Reviewer 2 Report

Abstract :

L8 : please precise UWT here for the first time you mention it.

L14-15 : mmol/L ?

Introduction :

L29 : I thin UW is enough as an abbreviation (because UWT could reprend UW time)

L33-35 : you could reword the sentence because of a lot of repetitions.

L53-58 : maybe here or somewhere else, you can mention that there is a paradox between the increase of heart rate during high intensity but also a bradychardia when you go underwater. So that’s why studying AS is more complicated with many physiological mechanisms are involved at the same time.

Materials and Methods

L91-102 : did you collect lactate measurements ? Do you are able to provide ventilatory thresholds ?

L108 : where was placed the OH1 please ?

Results

X

Discussion

L183-189 : those sentences aren’t necessary and are more appropriated for an introduction’s section.

L189-200 : all the results are detailed here again. So I think this is not appropriated. Juste recall the main findings and then discuss topic after topic, step by step.

I also imagine the possibility to test the relationships between VO2max and the parameters collected during the routine.

L202-226 : the paragraph seems a bit too long. I also imagine to see some comments on the applications on the field as for example the necessity to train under hypercapnia condition, which is different with hypoxic condition.

I think you have to mention the limits of your study and in particular, the fact that you have only 6 participants.

Author Response

We thank the reviewers for the time they have allocated to reading and commenting on the manuscript. The comments raised have improved the quality of manuscript and has identified potential opportunities for new lines of research. Below shows each of the reviewers' comments in bold and our replies. Any modifications in the manuscript have been highlighted in red. Any additional text also is seen below under each respective comment.

Abstract :

L8 : please precise UWT here for the first time you mention it.

‘Underwater ‘ has been abbreviated throughout to ‘UW’

And abbreviated version of under water has been included in the abstract section

L14-15 : mmol/L ?

Has been modified

Introduction :

L29 : I think UW is enough as an abbreviation (because UWT could reprend UW time)

We agree with the reviewer. Underwater has been abbreviated/modified from UWT to UW throughout the manuscript

L33-35 : you could reword the sentence because of a lot of repetitions.

This sentence has been reviewed and modified as requested by the reviewer  management and alone for changes

‘Anecdotally, athletes record greater scores if they are able to perform more movements in a routine allowing the opportunity to include more scoring opportunities into routine.’

L53-58 : maybe here or somewhere else, you can mention that there is a paradox between the increase of heart rate during high intensity but also a bradychardia when you go underwater. So that’s why studying AS is more complicated with many physiological mechanisms are involved at the same time.

The following text has been added. ‘This is further complicated by the bradycardia response observed during exercise performed in water’

Materials and Methods

L91-102 : did you collect lactate measurements ? Do you are able to provide ventilatory thresholds ?

Unfortunately blood lactate was not measured during the incremental test. Expired gasses were collected during the incremental test but we were primarily interested in maximal oxygen uptake and the differences/similarities in this variable between bike test and AS routine.

L108 : where was placed the OH1 please ?

The following text has been added:

‘around the mid portion of the bicep as recommended by the manufacturers..’

Results

X

Discussion

L183-189 : those sentences aren’t necessary and are more appropriated for an introduction’s section.

This text has been deleted

L189-200 : all the results are detailed here again. So I think this is not appropriated. Juste recall the main findings and then discuss topic after topic, step by step.

Please note comments for reviewer 1 who has recommended the opposite. The initial stages of the discussion section is to highlight the key results (which has been done). This is by no means an exhaustive listing of results. From this point on each of the main findings is discussed. The discussion section has been reviewed and we feel the composition of the discussion as it stands now is appropriate.

I also imagine the possibility to test the relationships between VO2max and the parameters collected during the routine.

We agree. This has been done

L202-226 : the paragraph seems a bit too long. I also imagine to see some comments on the applications on the field as for example the necessity to train under hypercapnia condition, which is different with hypoxic condition.

This paragraph has been reviewed and remains unchanged. The paragraph outlines firstly a key finding of the study and then addresses the potential application of the findings to training load monitoring and maladaptation to training.

I think you have to mention the limits of your study and in particular, the fact that you have only 6 participants.

Thank you for identifying this potential modification. Text has been added at the end of the discussion:

‘This is the first study in elite AS athletes to examine the relationship between performance-based parameters such as movement frequency and the metabolic responses to a simulated competitive AS routine. The subjects recruited for this investigation were small (n=6) which is a limitation of the study. However, these athletes represent an elite population. Future studies should aim to compare different subgroups of AS athletes (elite vs recreational). However, the complexity of the sport in terms of a choreographed performance make this difficult. This study shows the metabolic demands and has implications for long term training adaptation, this study did not qualify adaptations to UW exercise. This would be an interesting extension of this work that has implications for talent identification and optimising training in AS. A further limitation of the study is the nature of the AS routine which was a ‘simulated’ competition which may not necessarily reflect the demands of actual competition events.’

Round 2

Reviewer 1 Report

 The author’s explanation of the sample size is not satisfactory. The hypothesis cannot be tested if the sample size is less than recommended and there is a high chance of type II error.

More detail about the participant's recruitment is required. How many participants were invited? From where were the participants recruited? What are the inclusion/exclusion criteria? How many participants were excluded based on these criteria?- This is not answered properly

Author Response

More detail about the participant's recruitment is required. How many participants were invited? From where were the participants recruited? What are the inclusion/exclusion criteria? How many participants were excluded based on these criteria?- This is not answered properly

Thank you again for reviewing the manuscript

There were a total of 10 athletes in the entire squad. Remembering, this is an elite squad. All were invited to participate directly via the coach. However, the data collection occurred immediately before the Canada Games or an Olympic qualifier. n=2 were excluded based on illness.

In terms of inclusion/exclusion criteria. The criteria included "Be over 15 years of age and be free of injury and illness" which is why some were excluded.

We have added this criteria to the manuscript.

The criteria for inclusion were that participants’ were > 15 years of age and be free of injury and illness. Two athletes out of the squad of 10 were not included due to injury.

The author’s explanation of the sample size is not satisfactory. The hypothesis cannot be tested if the sample size is less than recommended and there is a high chance of type II error.

The reviewer assumes that use of parametric statistics in sport science is the appropriate method for analysis of data in these populations. In sport, especially elite sport, the margin of difference between success and failure is small. Distinguishing differences i.e. ‘significant’ differences with the use of paramedic statistics does not identify important and ‘meaningful’ effects in sport where the entire population has a small size anyway (For example see Bernards et al 2017) . Ideally, yes we would invite every nation to attend the testing , but this is not practical. We had the majority of the best performing AS athletes in the country - this is ‘the population ‘.

As we have previously stated , we could have engaged a lower level population but the participants in this study are elite and this poses immediate limitations of a study such as this where planning (choreography) is an important part of the preparation which takes months to perfect.

In addition, if we are to pursue a parametric approach to the statistics. There is very limited research on AS and so a valid sample size calculation is not possible. The small sample size is the best we could do given the level of competitive status the athletes at at - national level. 

References

Bernards, J.R. ,Sato, K., Haff, G.G. and Bazyler, C.D. (2017). Current Research and Statistical Practices in Sport Science and a Need for Change. Sports 5(4): 87

Reviewer 2 Report

The authors responded well to the reviewers.

Author Response

No reply required. Thank you reviewer 2